# Integrated Analysis of Transcriptome and Metabolome Reveals the Accumulation of Anthocyanins in Black Soybean (*Glycine max* L.) Seed Coats Induced by Low Nitrogen Concentration in the Nutrient Solution

**DOI:** 10.3390/plants14192993

**Published:** 2025-09-27

**Authors:** Suming Liang, Furong Si, Chenyang Guo, Yuan Chai, Tao Yang, Peng Wang

**Affiliations:** 1College of Agriculture, Shanxi Agricultural University, Taigu 030801, China; 13513632598@163.com (S.L.); 17835722724@163.com (F.S.); 18406552643@163.com (C.G.); 18810505137@163.com (Y.C.); yangtao18303425398@163.com (T.Y.); 2Shanxi Houji Laboratory, Shanxi Agricultural University, Taiyuan 030031, China

**Keywords:** black soybean, low nitrogen, anthocyanin, metabolites, differentially expressed genes

## Abstract

Anthocyanins are key antioxidants that play a significant role in plant responses to adverse stresses, including nitrogen deficiency. However, research on the metabolic and transcriptional regulation of anthocyanins in black soybean seed coats under low-nitrogen conditions remains limited. Here, we report that low-nitrogen treatment significantly alters the accumulation of anthocyanin metabolites and the gene expression profiles in black soybeans. Specifically, a greater number of differential anthocyanin metabolites are induced under low-nitrogen conditions, which contributes to the accumulation of anthocyanins in the seed coat. GO and KEGG enrichment analyses revealed that the differentially expressed genes (DEGs) are mainly enriched in multiple antioxidant pathways involved in responding to low-nitrogen stress; in flavonoid and phenylalanine metabolic pathways, as well as protein processing in endoplasmic reticulum, which are associated with anthocyanin biosynthesis; and in plant hormone signal transduction pathways involved in the regulation of anthocyanin accumulation. The expressions of genes encoding key enzymes in anthocyanin biosynthesis, such as dihydroflavonol 4-reductase (DFR) and O-methyltransferase (OMT), as well as genes encoding the blue light photoreceptor cryptochrome (CRY) and proteins related to cellular autophagy, were upregulated under low-nitrogen treatment. This suggests that these genes may play a key role in low-nitrogen-induced anthocyanin accumulation. This study provides a theoretical basis and novel perspective for understanding the regulatory mechanism underlying low-nitrogen-induced anthocyanin accumulation in black soybeans.

## 1. Introduction

Anthocyanins, a class of flavonoid secondary metabolites found in plants, are responsible for the pigmentation of plant organs and tissues [1]. Beyond being the most abundant group of flavonoid pigments in plants, anthocyanins have been widely recognized for their biological significance, including anti-inflammatory effects [2], anticancer properties [3], lipid-lowering effects [4], immunoregulatory functions [5], and the ability to reduce the risk of cardiovascular diseases [6].

Anthocyanins are synthesized in the cytoplasm from their precursor, phenylalanine (Phe), via the flavonoid metabolic pathway. After further processing in the endoplasmic reticulum, they are transported to the vacuoles [7]. The anthocyanin biosynthesis pathway involves multiple key enzymes. Among these, chalcone synthase (CHS), chalcone isomerase (CHI), flavanone 3-hydroxylase (F3H), flavonoid 3′-hydroxylase (F3′H), and flavonoid 3′,5′-hydroxylase (F3′5′H) catalyze the conversion of 4-coumaroyl-CoA into different types of dihydroflavonols. Subsequently, dihydroflavonol 4-reductase (DFR), anthocyanidin synthase (ANS), and UDPG-flavonoid glycosyl transferase (UFGT) are responsible for catalyzing the conversion of dihydroflavonols into various anthocyanins. Additionally, anthocyanin biosynthesis is tightly regulated by transcription factors. Genes belonging to the MYB, bHLH, and WD40 protein families can regulate the anthocyanin biosynthesis pathway either individually or by forming a ternary complex, thereby promoting anthocyanin accumulation [8,9]. For example, *MdMYB1* acts as a key regulator of anthocyanin biosynthesis and fruit coloration [10]. The expression level of the *Arabidopsis bHLH* transcription factor *GLABRA3* is positively correlated with anthocyanin accumulation [11]. In apples, MdbHLH3 binds to the promoters of the anthocyanin-biosynthetic structural genes *MdDFR* and *MdUFGT*, as well as the regulatory gene *MdMYB1*, to activate their expression [12]. In transient co-expression experiments, the combination of *MdbHLH3* and *MdMYB10* results in visible anthocyanin phenotypes in tobacco leaves [12,13].

Current evidence indicates that the synthesis and accumulation of anthocyanins in plants enhance their tolerance to both biotic and abiotic stresses, including pathogen invasion, low temperatures, drought, and high salinity [14,15]. When plants are exposed to high-light environments, anthocyanins in their leaves accumulate in large quantities to absorb excess light energy. This accumulation helps resist ultraviolet radiation, mitigate adverse effects on photosynthesis, and protect chlorophyll, proteins, and DNA from damage [16]. Under low-temperature stress, anthocyanins function in scavenging reactive oxygen species (ROS) to reduce plant damage. They also regulate chlorophyll metabolism and photosynthesis-associated genes, thereby inhibiting chlorophyll degradation and preserving photosynthetic capacity [17,18]. When subjected to drought or salt-induced osmotic stress, anthocyanins accumulate in large quantities and act as osmotic regulators to maintain water balance and stabilize the osmotic pressure of plant cells [19]. Heavy metal stress impairs the root system’s capacity to take up water and nutrients, damages photosynthetic machinery, and consequently constrains plant growth. As non-enzymatic antioxidants and metal chelators, anthocyanins protect plant cells against oxidative damage, facilitate ionic homeostasis, enhance the sustained acquisition of water and nutrients, and ensure the normal ontogenesis of plants [20].

Nitrogen is one of the essential macronutrients for plant growth [21]. Nitrogen deficiency, as an abiotic stress for plants, often acts as a limiting factor for plant growth [22]. Similarly to various abiotic stresses, nitrogen deficiency has highly adverse impacts on diverse metabolic activities in plants, leading to the accumulation of ROS and the associated oxidative stress. During low-nitrogen-induced leaf senescence, the accumulation of anthocyanins can minimize stress-related oxidative damage and facilitate nutrient remobilization from older leaves to younger active tissues [23]. In general, anthocyanin synthesis is upregulated under low-nitrogen conditions, whereas it is downregulated under high-nitrogen conditions [24,25,26]. For instance, the anthocyanin content in tomato leaves increases under nitrogen-deficient conditions, with concurrent increases in the transcription levels of *CHS* and *DFR* [27]. Low-nitrogen stress induces anthocyanin synthesis in detached crabapple flowers, causing the color of the explants to change from green to red, and the expression level of *ANS* is significantly increased [28]. During the suspension cell culture of *Arabidopsis thaliana*, reducing the nitrate nitrogen content induces the synthesis and accumulation of anthocyanins in the cells [29]. Studies on *Arabidopsis thaliana* in low-nitrogen environments have shown that high anthocyanin accumulation helps plants cope with low-nitrogen stress by enabling the efficient utilization of nitrate metabolism [30].

Moreover, numerous studies have focused on the relationship between growth hormones and anthocyanins in plants under nitrogen stress. In *Brassica oleracea*, the expression levels of multiple genes involved in abscisic acid (ABA) biosynthesis are more significantly upregulated in purple leaves than in green leaves, suggesting that ABA may promote the intense purple pigment deposition in the inner purple leaves [31]. Transcriptome analysis of mature and unripe ‘Hongyu’ apples identified DEGs involved in the biosynthesis pathways of ethylene, ABA, auxin, gibberellin, and anthocyanin [32]. Additionally, jasmonate (JA) treatment can induce anthocyanin biosynthesis, a process regulated by the *MYB9* and *MYB11* transcription factors [33]. In *Jatropha curcas* L., the JA biosynthesis pathway is enhanced under nitrogen-deficient conditions, which can affect anthocyanin accumulation. Concurrently, several genes in the *MYB* and *WRKY* families associated with anthocyanin synthesis are upregulated [34].

Black soybean (*Glycine max* (L.) Merr.) is a nutritious legume [35]. In China, it is mainly distributed in mountainous regions and areas with poor soil quality in the north, where traditional agricultural production methods are still employed. As a result, black soybean production is often affected by soil nutrient deficiencies. Additionally, environmental damage caused by excessive fertilizer application has attracted widespread attention. Reducing nitrogen fertilizer input and improving nitrogen use efficiency have become important requirements for the development of green agriculture. Black soybeans are widely recognized as a food with medicinal value due to their high content of bioactive components such as anthocyanins [36]. The anthocyanin content in their seeds is a key indicator determining the nutritional and economic value of black soybeans. Studies have shown that environmental factors significantly affect anthocyanin accumulation, with light, temperature, water, sugars, hormones, phosphorus, and nitrogen all involved in regulating anthocyanin synthesis [15]. Therefore, research on the impact of a low-nitrogen environment on anthocyanin accumulation in black soybean seeds is of great significance.

Here, we conducted an untargeted metabolomics analysis to investigate the accumulation patterns of anthocyanin metabolites in the seed coat of black soybeans under nitrogen-deficient conditions. Additionally, we analyzed the molecular mechanisms underlying low-nitrogen-induced anthocyanin accumulation at the transcriptional level. Finally, we identified some candidate genes and proposed a potential regulatory pathway involved in anthocyanin accumulation in the black soybean seed coat under low-nitrogen treatment. However, the specific regulatory mechanism underlying low-nitrogen-induced anthocyanin accumulation requires further investigation.

## 2. Results

### 2.1. Effect of Low-Nitrogen Stress on Anthocyanin Content in Black Soybeans

As shown in Figure 1a, the seed coat color gradually changed from green to black from the SG (stage when seeds swell to fill the pod cavity and the seed coat is green, 29 days after flowering, 29 DAF) to SB stages (stage when the seed coat has fully turned black, 53 DAF), accompanied by anthocyanin accumulation. At the SG stage, the anthocyanin content in the seed coat was relatively low, with no significant differences observed between the different nitrogen levels (Figure 1b, Appendix A). However, the anthocyanin content increased significantly at both the SC and SB stages. Notably, at the SB stage, the anthocyanin content under the LN (low nitrogen stress) treatment was significantly higher than that under the NN (normal nitrogen application) treatment. These results indicate that low-nitrogen stress can promote anthocyanin accumulation in the seed coat of black soybeans.

### 2.2. Analysis of Anthocyanin Metabolites in Samples

Based on the results of principal component analysis (PCA) (Figure 2a, Appendix A), samples from different seed developmental stages (SG, SC, SB) were clearly separated, indicating significant differences in anthocyanin metabolites in the seed coat of black soybeans across various developmental stages.

A total of 34 anthocyanin metabolites were detected in all seed coat samples, including callistephin (1), cyanidins (9), delphinidins (7), epicatechin (1), malvidins (3), pelargonidin (1), peonidins (3), petunidins (8), and procyanidin B2 (1) (Appendix A). The proportion of peonidin content increased significantly as the seed coat began to turn from green to black (SC and SB stages), whereas the proportions of malvidin and petunidin in the LN treatment were significantly higher than those in the NN treatment at the SB stage (Figure 2b, Appendix A).

As shown in the clustering analysis results (Figure 2c, Appendix A), samples from the SC and SG stages clustered together, whereas those from the SB stage formed a distinct cluster. Additionally, samples subjected to different nitrogen levels within the same stage were clearly grouped into two distinct clusters, indicating that low-nitrogen stress exerted a significant impact on the accumulation of anthocyanin metabolites in the black soybean seed coat. These metabolites were divided into three groups via cluster analysis (Figure 2c). The first group contained 23 metabolites that accumulated gradually from the SG stage to the SB stage. The second group consisted of 4 metabolites whose content first increased and then decreased. The third group included 7 metabolites with higher content at the SG stage, which were subsequently down-regulated.

### 2.3. Identification of Differentially Anthocyanin Metabolites

Anthocyanin differential metabolites were screened based on the criteria of *p* < 0.05 and VIP > 1.0 (“Variable Importance in Projection”, which denotes the extent to which each metabolite contributes to the observed differences). As shown in Figure 3 and Appendix A, a total of 13 differential metabolites were identified between the low-nitrogen (LN) and normal-nitrogen (NN) treatments at the SG stage (LNSG vs. NNSG), among which 7 were upregulated and 6 were downregulated; 7 differential metabolites were identified between the LN and NN treatments at the SC stage (LNSC vs. NNSC), among which 5 were upregulated and 2 were downregulated; and 8 differential metabolites were identified between the LN and NN treatments at the SB stage (LNSB vs. NNSB), among which 6 were upregulated and 2 were downregulated. These results indicated that the number of upregulated differential anthocyanin metabolites in LN treatment exceeds that of the downregulated ones, ultimately leading to a higher anthocyanin content in the LN treatment.

### 2.4. Analysis of Differential Expressed Genes in Seed Coat

Based on transcriptome sequencing results, DESeq2 was employed to analyze differential expression at both the transcriptional and gene levels. DEGs were filtered using the criteria of |log_2_FC| > 1 and FDR < 0.05.

To understand the relationships among the samples, hierarchical clustering was performed on the gene expression matrix (Figure 4a, Appendix A). All 18 samples were divided into two clusters: one cluster contained samples from the SB stage, and the other cluster aggregated samples from the SG and SC stages. Furthermore, distinct expression patterns of DEGs were also observed between the LN and NN treatments at various stages (Appendix A). These transcriptomes showed good biological repeatability under the same conditions, and the expression profiles of DEGs exhibited stable and unique responses to different nitrogen treatments.

Further analysis of DEGs across different nitrogen treatments (Figure 4b–e, Appendix A) revealed the following: a total of 92 DEGs were identified between the LN and NN treatments at the SG stage (LNSG vs. NNSG), of which 53 were upregulated and 39 were downregulated; 137 DEGs were identified between the LN and NN treatments at the SB stage (LNSB vs. NNSB), with 53 upregulated and 84 downregulated; and 1031 DEGs were identified between the LN and NN treatments at the SC stage (LNSC vs. NNSC), including 158 upregulated and 873 downregulated genes.

### 2.5. Enrichment Analysis of Differentially Expressed Genes

To deduce the potential functions of these DEGs, GO and KEGG enrichment analyses were performed between the LN and NN treatments at various stages, and the sequence annotations of these genes were characterized.

Specifically, among the 92 DEGs in the SG stage, 43 DEGs (46.74%) were assigned to 875 GO terms. As shown in Figure 5a and Appendix A, the DEGs were mainly concentrated in the following categories: for cellular component (CC), plant-type vacuole membrane (GO:0009705) and plant-type vacuole (GO:0000325); for molecular function (MF), monoatomic ion transmembrane transporter activity (GO:0015075), transmembrane transporter activity (GO:0022857), and transporter activity (GO:0005215); and for biological process (BP), transmembrane transport (GO:0055085), transport (GO:0006810), monoatomic ion transmembrane transport (GO:0034220), and monoatomic ion transport (GO:0006811). Furthermore, the KEGG pathway enrichment was mainly manifested in protein processing in endoplasmic reticulum (6 genes, all downregulated), glycolysis/gluconeogenesis (4 genes, 2 downregulated and 2 upregulated), and plant-pathogen interaction (5 genes, 2 downregulated and 3 upregulated) (Figure 5b and Appendix A).

During the SC stage, among the 1031 DEGs, 545 DEGs (52.86%) were assigned to 2611 GO terms. As shown in Figure 6a and Appendix A, the DEGs were mainly concentrated in the following categories: for cellular component (CC), cell periphery (GO:0071944) and plasma membrane (GO:0005886); for molecular function (MF), transferase activity (GO:0016740), catalytic activity acting on a protein (GO:0140096), DNA-binding transcription factor activity (GO:0003700), and transcription regulator activity (GO:0140110); and for biological process (BP), response to stimulus (GO:0050896), response to stress (GO:0006950), biological regulation (GO:0065007), and regulation of biological process (GO:0050789). Furthermore, the KEGG pathway enrichment was mainly manifested in plant-pathogen interaction (48 genes, 45 downregulated and 3 upregulated), plant hormone signal transduction (25 genes, 22 downregulated and 3 upregulated), MAPK signaling pathway-plant (13 genes, all downregulated), and biosynthesis of various plant secondary metabolites (7 genes, 4 downregulated and 3 upregulated) (Figure 6b and Appendix A).

Furthermore, among the 137 DEGs in the SB stage, 85 DEGs (62.04%) were assigned to 1095 GO terms. As shown in Figure 7a and Appendix A, the DEGs were mainly concentrated in the following categories: for cellular component (CC), motile cilium (GO:0031514) and sperm flagellum (GO:0036126); for molecular function (MF), omega peptidase activity (GO:0008242), gamma-glutamyl-peptidase activity (GO:0034722), and Hsp90 protein binding (GO:0051879); and for biological process (BP), response to oxidative stress (GO:0006979), response to reactive oxygen species (GO:0000302), and response to temperature stimulus (GO:0009266). Additionally, KEGG pathway enrichment was mainly observed in protein processing in endoplasmic reticulum (21 genes, all upregulated), cyanoamino acid metabolism (3 genes, 2 downregulated and 1 upregulated), and plant-pathogen interaction (4 genes, 1 downregulated and 3 upregulated) (Figure 7b and Appendix A).

### 2.6. Integrated Transcriptomic and Metabolomic Analysis

An integrated analysis was performed on GO annotations and KEGG pathways associated with flavonoid and anthocyanin synthesis, with the aim of identifying candidate genes probably responsible for inducing anthocyanin accumulation under low nitrogen (LN) stress. As shown in Table 1, 12 genes with significant changes (|log_2_FC| > 1 and FDR < 0.05) in their expression levels were selected.

*Glyma.13G309300* encodes an arogenate dehydratase (ADT) that catalyzes the conversion of aromatic acids to phenylalanine, playing a key role in phenylalanine biosynthesis. *Glyma.20G114200* encodes cinnamic acid-4-hydroxylase (C4H), a key enzyme in the phenylalanine metabolic pathway that catalyzes the conversion of cinnamic acid to coumaric acid. *Glyma.13G089200* is a blue-light photoreceptor cryptochrome (CRY) that likely functions in anthocyanin biosynthesis. *Glyma.17G171100* and *Glyma.14G200900* are annotated as O-methyltransferases (OMTs), which catalyze the oxymethylation of plant secondary metabolites. *Glyma.08G109500* and *Glyma.17G173200* are annotated as chalcone synthase (CHS) and dihydroflavonol reductase (DFR), respectively, both of which are key enzymes in the anthocyanin synthesis pathway. *Glyma.09G204500* and *Glyma.08G271900* belong to the *MYC* transcription factor family, which is involved in regulating jasmonic acid signaling-mediated anthocyanin synthesis. They can also directly affect structural genes in the anthocyanin pathway through both activation and inhibition.

### 2.7. Verification of DEGs Using qRT-PCR

The expression levels of 12 candidate genes involved in anthocyanin synthesis were analyzed using specific primers. The results showed that the changing trends of gene expression levels were consistent with those of the transcriptome data. As revealed by qRT-PCR (Figure 8 and Appendix A), at the SC stage, the expression levels of *Glyma.17G173200*, *Glyma.14G200900*, and *Glyma.17G171100* were significantly higher in the LN treatment than in the NN treatment. Additionally, *Glyma.13G089200* was upregulated in the LN treatment at the SB stage. In contrast, the expression levels of the remaining genes were significantly lower in the LN treatment compared with the NN treatment.

## 3. Discussion

In general, anthocyanin synthesis gradually increases during the color transition process in anthocyanin-rich fruits such as grapes, *Lycium chinense*, blood oranges, and apples. Furthermore, anthocyanin synthesis is induced in response to stress, enabling it to counteract the adverse effects of various stressors [37]. The results of the present study are consistent with these findings. Anthocyanin content increased substantially as the black soybean seed coat began to transition from green to black (SC stage), and at the stage when the seed coat had completely turned black (SB stage), the content was significantly higher under LN treatment than under NN treatment. These results demonstrate that lower nitrogen levels facilitate anthocyanin accumulation in the seed coat, suggesting a potential strategy to enhance the nutritional value of black soybeans under reduced nitrogen fertilizer application. It is worth noting that the anthocyanin content was significantly higher in the NN treatment than in the LN treatment at the SC stage. It is hypothesized that this result may be attributed to the fact that anthocyanins exert their antioxidant effects to counteract the oxidative damage imposed on plants by low-nitrogen stress.

In general, anthocyanins undergo modifications by glycosyl, aromatic, or aliphatic acyl moieties, giving rise to hundreds of anthocyanin molecules with diverse colors and stabilities [38]. Studies have reported that at least 27 categories and over 550 types of anthocyanins have been identified, among which pelargonidin, cyanidin, delphinidin, peonidin, petunidin, and malvidin account for approximately 92% of naturally occurring anthocyanins [39,40,41]. To further investigate the accumulation patterns of various anthocyanin metabolites under LN treatment, an untargeted metabolomic analysis of anthocyanins in seed coat samples was conducted at different stages of seed development. In this experiment, a total of 34 anthocyanin metabolites belonging to 9 categories were detected across all samples. The proportions of petunidin and malvidin were relatively high at the SG stage, whereas the proportion of peonidin increased significantly during the SC and SB stages. These results indicated that peonidin accounted for the highest proportion among anthocyanin metabolites in the seed coat of this black soybean variety. Cluster analysis of samples and metabolites revealed significant differences in the expression profiles of anthocyanin metabolites between samples at the SB stage and those at the SC and SG stages, suggesting that both the types and contents of anthocyanin metabolites play a crucial role in determining the seed coat color of black soybeans. Additionally, samples from the same seed development stage were divided into two groups (LN and NN), indicating that LN treatment exerts a significant effect on the expression of anthocyanin metabolites in the seed coat of black soybeans. Furthermore, the number of differential anthocyanin metabolites that upregulated in LN treatment was greater than the number of downregulated. These results fully demonstrate the role of low nitrogen in inducing anthocyanin accumulation in the seed coat of black soybeans.

Transcriptome sequencing (RNA-seq) has been widely used to identify key genes and pathways involved in plant responses to environmental changes [42,43]. However, omics research on the impact of nitrogen treatment on anthocyanin accumulation remains relatively limited, and the relevant regulatory pathways have not yet been fully elucidated. To investigate the transcriptional regulatory mechanisms underlying low-nitrogen-induced anthocyanin accumulation, this study analyzed the transcriptome of black soybean seed coats at various seed development stages under low-nitrogen treatment. The results revealed that the number of DEGs during the SG stage was the smallest, suggesting that the transcriptome profile under LN treatment was similar to that under NN treatment. These DEGs were mainly enriched in processes such as monatomic ion transport, monatomic ion homeostasis, and transmembrane transporter activity, as well as in components including plant-type vacuoles and vacuolar membranes. At this stage, the impact of low-nitrogen stress was relatively mild, exerting only minor effects on the homeostasis of the intracellular environment. A total of 1031 DEGs were identified during the SC stage, representing the largest number observed across the three stages. Among these, 158 were upregulated and 873 were downregulated, suggesting that LN treatment exerts the most significant impact on the transcriptional profile of black soybean seeds at this stage. Results from GO enrichment analysis indicated that these DEGs are primarily enriched in multiple biological processes, including protein kinase activity, transferase activity, response to stimulus, response to external stimulus, and response to hormones. At this point, a large number of genes from various metabolic pathways need to be mobilized to participate in the response to low-nitrogen stress. During the SB stage, a total of 137 DEGs were identified, among which 53 were upregulated and 84 were downregulated. These DEGs are mainly enriched in processes such as response to hydrogen peroxide, response to oxidative stress, and response to reactive oxygen species (ROS). At this stage, the impact of low-nitrogen stress on plants primarily manifests as peroxidative damage induced by ROS, and plants thus rely on various antioxidant pathways to cope with the stress.

KEGG enrichment pathway analysis revealed that DEGs were mainly concentrated in protein processing in the endoplasmic reticulum, plant-pathogen interaction, phenylalanine biosynthesis, MAPK signaling pathway, plant hormone signal transduction, flavonoid biosynthesis, and glycolysis/gluconeogenesis. Studies have indicated that these pathways may be involved in low-nitrogen response and anthocyanin accumulation. When plants are subjected to stress, they typically regulate the expression of related genes through plant-pathogen interactions, plant hormone signaling, and MAPK signaling pathways [44]. In plants, anthocyanins are synthesized from phenolic precursors via the phenylpropanoid pathway in the endoplasmic reticulum [45]. Under stress conditions, plant hormones such as abscisic acid and jasmonic acid also act synergistically with anthocyanins, promoting their accumulation to enhance stress resistance [26]. Although anthocyanin accumulation is not directly associated with energy synthesis metabolism, energy derived from the glycolysis/gluconeogenesis and oxidative phosphorylation pathways can also function as signaling molecules to activate genes related to anthocyanin synthesis [46]. Furthermore, during the color transition phase, the DEGs were enriched in pathways involved in flavonoid and phenylalanine biosynthesis, suggesting that low nitrogen may directly affect phenylalanine synthesis and thereby influence anthocyanin production.

Furthermore, during the SB stage, certain DEGs were enriched in pathways related to the induction of programmed cell death and the regulation of autophagy. It is well known that nutrient recycling and remobilization processes are crucial for plants under nitrogen deficiency [47]. Autophagy represents one of the most important pathways that facilitate nutrient recycling by degrading unwanted or damaged organelles, proteins, and cytoplasmic components [48]. Previous studies have highlighted the significance of autophagy in regulating nitrogen remobilization under low-nitrogen conditions [49,50]. Additionally, several studies have elaborated on the role of autophagy in anthocyanin accumulation [51]. Arabidopsis *atg* mutants, which display hypersensitivity to nitrogen deficiency, exhibit less efficient nitrogen remobilization and reduced anthocyanin accumulation [52,53]. Thus, We hypothesize that low nitrogen enhances autophagic activity, and the upregulated expression of autophagy-related genes subsequently supports anthocyanin biosynthesis. This represents a potential regulatory pathway underlying low-nitrogen-induced anthocyanin accumulation in the black soybean seed coat, with the specific molecular mechanisms requiring further investigation.

Further screening and analysis were conducted on key genes involved in GO and KEGG enriched pathways. Based on gene annotations, several candidate genes potentially closely associated with low-nitrogen-induced anthocyanin accumulation were identified. *Glyma.17G173200* encodes DFR, a key enzyme in anthocyanin biosynthesis that regulates anthocyanin synthesis [54]. *Glyma.17G171100* and *Glyma.14G200900* encode OMT, which can methylate oxygen atoms in plant secondary metabolites. OMTs are important genes involved in anthocyanin biosynthesis, as glycosylation, acylation, and methylation of aglycone structures are critical steps in the formation of anthocyanins (e.g., cyanidin and pelargonidin derivatives) [55,56]. *Glyma.13G089200* encodes the blue light photoreceptor CRY, which plays a crucial role in various developmental processes [57,58]. Overexpression of *PagCRY1* dramatically increased anthocyanin accumulation in *Populus* [59]. The expression levels of these genes were significantly upregulated in the seed coat of black soybeans, suggesting that they may be induced by low-nitrogen stress to participate in anthocyanin synthesis and accumulation.

## 4. Materials and Methods

### 4.1. Plant Materials

The new black soybean variety Jinda Heidou No. 6 (JH6), characterized by a black seed coat and green cotyledons, was selected as the experimental material. In 2023, JH6 plants were cultivated in pots (37 cm in diameter, 34 cm in height) within a greenhouse at Shanxi Agricultural University. Each pot was filled with 8 kg of a mixed substrate composed of vermiculite and sand in a 2:1 volume ratio, and 10 seeds were sown per pot. After germination, 5 seedlings with consistent growth status were selected from each pot.

### 4.2. Low Nitrogen Stress

Plants used in this study were irrigated with “modified Hoagland nutrient solution (nitrogen- and Ca(NO_3_)_2_-free)” (Qingdao Hope Bio-Technology Co., Ltd., Qingdao, China, Product Code: HB8870-9), which contains KCl (152.828 mg/L), K_2_HPO_4_·3H_2_O (456.638 mg/L), MgSO_4_·7H_2_O (246.48 mg/L), CHFeN_2_NaO_8_ (NaFeEDTA) (11.17 mg/L), H_3_BO_3_ (1.546 mg/L), MnSO_4_·H_2_O (0.338 mg/L), CuSO_4_ (0.125 mg/L), ZnSO_4_·H_2_O (0.576 mg/L), and Na_2_MoO_4_ (0.102 mg/L). Since this nutrient solution lacks a nitrogen source, we added ammonium nitrate (NH_4_NO_3_) as the nitrogen source in the experiment and established different nitrogen levels. Prior to the peak flowering stage, JH6 plants were irrigated with 1/2 Hoagland nutrient solution at a nitrogen concentration of 7.5 mmol/L. After the peak flowering stage, two nitrogen treatments were implemented: low nitrogen stress (LN) with a nitrogen concentration of 1.5 mmol/L, and normal nitrogen application (NN) with a nitrogen concentration of 7.5 mmol/L. We irrigated the plants with the nutrient solution every 3 days, at a volume of 500 mL per application.

Subsequently, seed coat samples were collected at three distinct seed developmental stages: 29 days after flowering (DAF), when seeds swell to fill the pod cavity and the seed coat is green (designated as the SG stage); 45 DAF, when the seed coat begins to turn from green to black (designated as the SC stage); and 53 DAF, when the seed coat has fully turned black (designated as the SB stage). All samples were immediately frozen in liquid nitrogen and then stored at −80 °C for subsequent determinations of anthocyanin content, as well as transcriptome and metabolome analyses.

### 4.3. Determination of Anthocyanin Content

Anthocyanins were extracted from seed coat samples (approximately 0.1 g each) using 1 mL of 5% formic acid. The mixture was centrifuged at 12,000 rpm for 20 min at 4 °C (Dragon Lab centrifuge, D1524R, rotor model AS24−2). This process was repeated several times until the supernatant became colorless. The absorbance of the supernatant was measured at 530 nm, with 5% formic acid used as the blank control (Thermo Fisher Scientific Inc., Waltham, MA, USA; Model: Multiskan GO).

A 100 μg/mL cyanidin-3-O-sophoroside (Sigma-Aldrich Corporation, St. Louis, MO, USA, CAS No. 38820−68−7) solution was used as the standard. Standard solutions of various concentrations were then prepared according to Appendix A, and a standard curve was plotted (Appendix A). The anthocyanin content in the samples was calculated using the following formula: Anthocyanin content (μg/g) = C × V ÷ M; C: concentration derived from the standard curve (μg/mL); V: total volume of the extract (mL); M: sample weight (g).

### 4.4. RNA Extraction, Library Construction, and Sequencing

Samples were sent to Jiangsu Sanshu Biotechnology Co., Ltd. (Nantong, China) for transcriptome sequencing. The experimental procedure is briefly described as follows: mRNA was enriched from total RNA using magnetic beads coated with Oligo (dT) (RNAprep Pure Polysaccharide & Polyphenol Plant Total RNA Extraction Kit, Centrifugal Column Type, Catalog No. DP441) (TIANGEN Biotechnology Co., Ltd., Beijing, China). Subsequently, the enriched mRNA was randomly fragmented with fragmentation buffer. Using the fragmented mRNA as a template, double-stranded cDNA was synthesized via reverse transcriptase, followed by terminal repair with an End Repair Mix (ABclonal Biotechnology Co., Ltd., Woburn, MA, USA) and addition of an adenine (A) base to the 3′ end to form a “Y”-shaped adapter. Library enrichment was performed by PCR amplification for 15 cycles (Fast RNA-seq Lib Prep Kit V2, Cat. No. RK20306) (ABclonal Biotechnology Co., Ltd., Woburn, MA, USA), after which sequencing was carried out using a second-generation high-throughput sequencing platform (DNBSEQ-T7, Shenzhen MGI Tech Co., Ltd., Shenzhen, China).

### 4.5. Transcriptomic Analysis

Raw sequencing data were filtered using fastp software (v0.23.4) to generate high-quality sequencing data (clean data), the parameters were set as follows: length filtering (−l 50); quality trimming (−n 5, −q 20); maximum read length limit (−M 20); and parallel processing (−W 4). The STAR software (v2.7.11a) was then used to map the clean data to the soybean reference genome (*Glycine max* v2.1) via sequence alignment (with default parameters; the command was: STAR–genomeDir genome_index–readFilesIn reads.fq.). Transcriptome data quality was evaluated using RSeQC software (version 5.0.1) with default parameters (Appendix A). FPKM values were used as indicators to measure the expression levels of genes/transcripts in samples, followed by sample correlation analysis and principal component analysis based on these FPKM values. DESeq2 was employed for differential expression analysis at both transcriptional and gene levels. DEGs were screened using the criteria of |log_2_FC| > 1 and FDR < 0.05. GO and KEGG pathway enrichment analyses of DEGs were performed using the Python SciPy toolkit (v1.14.1). Comparative analyses were conducted in the following groups: (1) SG stage: LNSG vs. NNSG; (2) SC stage: LNSC vs. NNSC; (3) SB stage: LNSB vs. NNSB.

Significantly enriched GO functional categories of DEGs were identified using Fisher’s exact test (*p* < 0.05) and analyzed according to cellular component (CC), molecular function (MF), and biological process (BP). Furthermore, the top 10 significantly enriched GO categories were selected for display in a bar chart. If there were fewer than 10 categories, all of them would be displayed. The KEGG enrichment was evaluated using the rich factor (GeneRatio, the ratio of the number of DEGs annotated to a KEGG pathway to the total number of DEGs) and *p*-value: the higher the ratio, the greater the degree of enrichment of DEGs in that KEGG pathway; the closer the *p*-value is to zero, the more significant the enrichment. The top 20 significantly enriched KEGG pathways were selected for display; if there were fewer than 20, the top 20 pathways based on significance ranking would be selected.

### 4.6. Metabolomics Analysis

The materials used for metabolomics analysis were identical to those for transcriptome sequencing. An appropriate amount of each sample was extracted twice with 1 mL of a methanol:water:formic acid mixture (70:30:1, *v*/*v*/*v*) to ensure complete dissolution of anthocyanins. The extracted samples were analyzed using a UPLC-Orbitrap-MS system (Thermo Fisher Scientific Inc., Waltham, MA, USA). High-resolution mass spectrometry (HRMS) data were acquired with a Q Exactive hybrid Q-Orbitrap mass spectrometer (Thermo Fisher Scientific Inc., Waltham, MA, USA) equipped with a heated electrospray ionization (ESI) source, using the Fullscan-MS^2^ acquisition method. Raw MS data were collected on the Q-Exactive instrument via Xcalibur 4.1 software (Thermo Scientific) and processed using Progenesis QI (Waters Corporation, Milford, CT, USA). Metabolites with a *p*-value < 0.05 and VIP > 1, identified through *t*-test combined with the multivariate analysis OPLS-DA, were designated as differential metabolites.

### 4.7. Gene Expression Analysis

Total RNA was extracted from the seed coats of five individual seeds per treatment at different stages using a TRIzol kit, following the manufacturer’s instructions. After treatment with DNase I (Takara Bio. Inc., Kusatsu, Japan), 2 μg of total RNA was reverse-transcribed using the FastQuant RT kit (Tiangen Biotech. Co., Ltd., Beijing, China). Real-time quantitative PCR was performed with 2 × Q3 SYBR qPCR Master Mix (Tolobio Co., Ltd., Nanjing, China), using CYP2 as the reference gene, with three biological replicates. Specific primers were designed using online tools provided by the National Center for Biotechnology Information (NCBI) (Appendix A). Relative gene expression levels were calculated using the 2^−ΔΔCt^ method [60].

### 4.8. Statistical Analysis

The obtained data were analyzed using IBM SPSS Statistics 20. Significant differences between the means (averages of at least three replicates) were determined via Duncan’s multiple range test at the *p* < 0.05 level. All figures were generated using GraphPad Prism 7.

## 5. Conclusions

In summary, a total of 34 anthocyanin metabolites were detected in all seed coat samples. Among these, a greater number of anthocyanin metabolites were induced by low-nitrogen treatment, ultimately contributing to a higher anthocyanin content in the seed coat of black soybean cultivar JH6. Low-nitrogen treatment significantly altered the gene expression profiles across different stages of seed development. The differentially expressed genes identified under low-nitrogen treatment primarily mediate JH6′s response to low-nitrogen stress and anthocyanin accumulation via pathways including stress response, flavonoid and phenylalanine metabolism, and plant hormone signal transduction. Notably, the expression levels of genes encoding DFR, OMT, CRY, and proteins related to cellular autophagy were upregulated, suggesting that these genes may be involved in the synthesis and accumulation of anthocyanins under low-nitrogen conditions.

## Figures and Tables

**Figure 1 plants-14-02993-f001:**
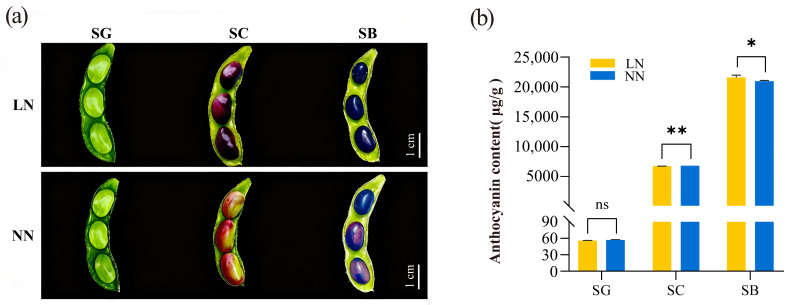
Effect of low-nitrogen treatment on anthocyanin content in the seed coat of black soybean. LN, low nitrogen stress (with a nitrogen concentration of 1.5 mmol/L). NN, normal nitrogen application (with a nitrogen concentration of 7.5 mmol/L). SG, stage when seeds swell to fill the pod cavity and the seed coat is green (29 DAF). SC, stage when the seed coat begins to turn from green to black (45 DAF). SB, stage when the seed coat has fully turned black (53 DAF). (**a**) Changes in seed coat color of black soybeans under different treatments at various developmental stages. The length of the scale bar represents an actual length of 1 cm. (**b**) Anthocyanin content in the seed coat of black soybeans under different treatments at various developmental stages. Error bars represent the standard deviation (SD) of three biological replicates. Significance analysis was performed on the data of LN and NN treatments at each developmental stages. *, *p* < 0.05. **, *p* < 0.01. ns (no significant difference), *p* > 0.05.

**Figure 2 plants-14-02993-f002:**
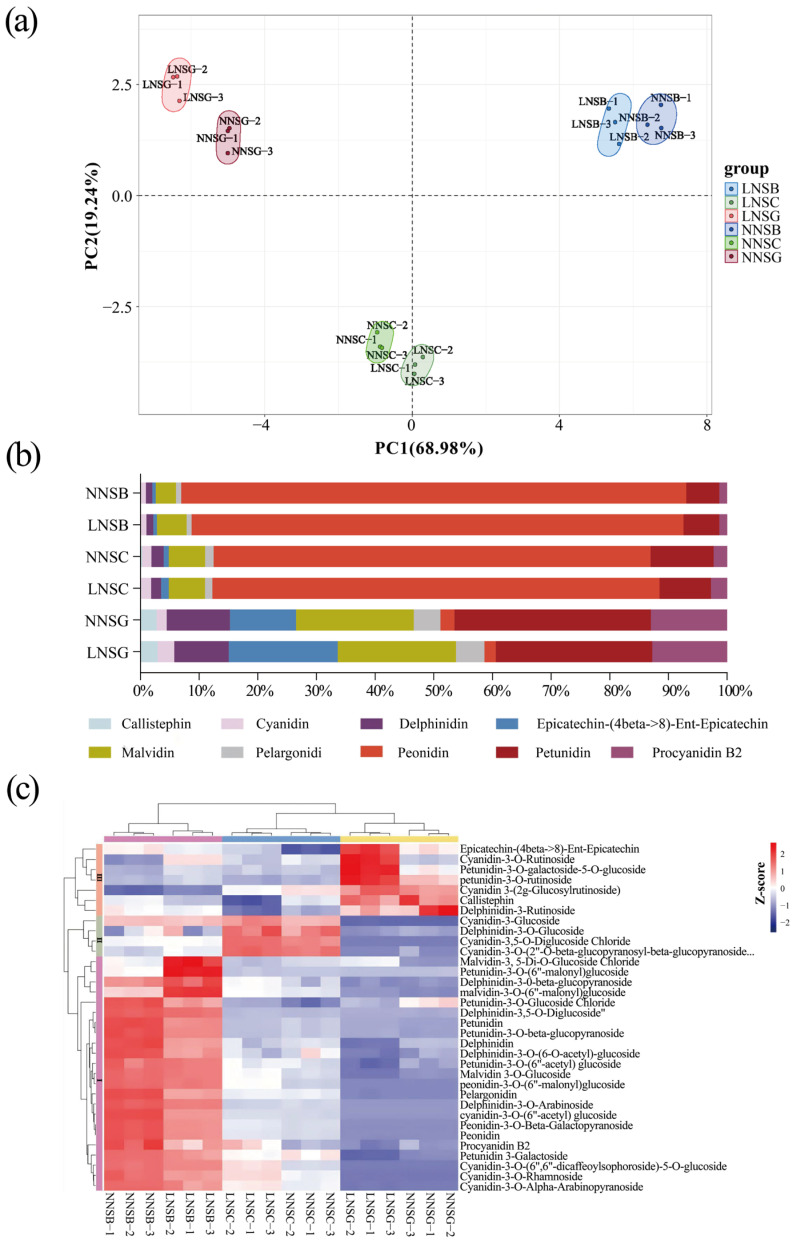
Analysis of anthocyanin metabolites in samples. (**a**) Principal component analysis (PCA) of the samples. (**b**) Relative contents of different anthocyanin metabolites across various developmental stages. (**c**) Cluster heatmap of samples and metabolites. The color scale, ranging from blue (−2) to red (2), represents the Z-score. The darker the blue, the lower the metabolite expression level; the darker the red, the higher the metabolite expression level.

**Figure 3 plants-14-02993-f003:**
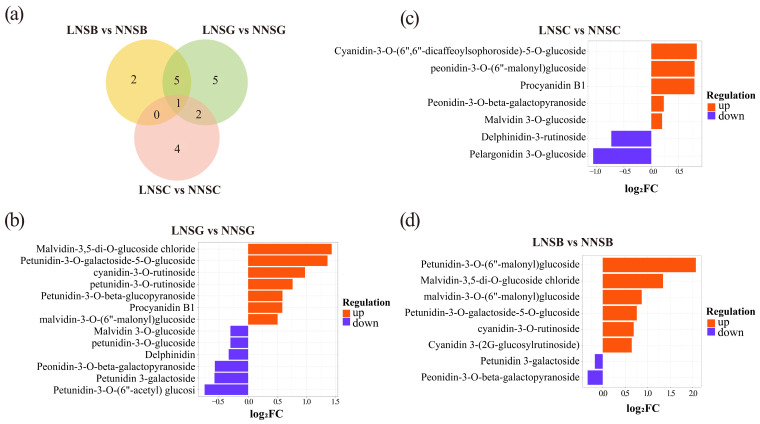
Identification of differential anthocyanin metabolites. (**a**) Venn diagram of differential anthocyanin metabolites; (**b**–**d**) Differential anthocyanin metabolites at the SG (**b**), SC (**c**) and SB (**d**) stages.

**Figure 4 plants-14-02993-f004:**
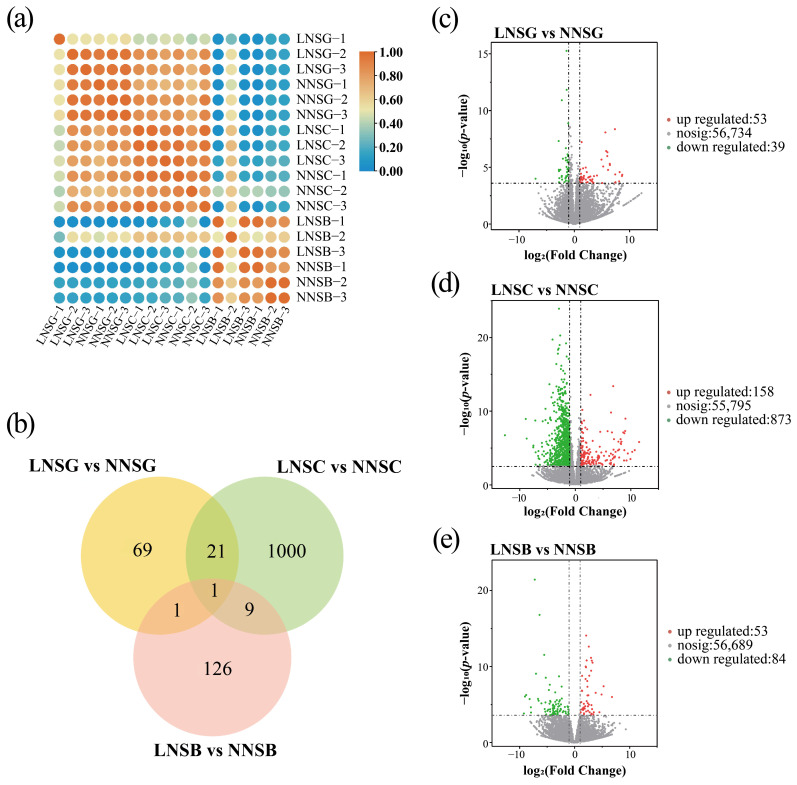
Analysis of differentially expression genes. (**a**) Cluster heatmap of samples; The color scale refers to the Pearson correlation coefficient. (**b**) Venn diagram of DEGs. (**c**–**e**) Volcano plots of DEGs in comparisons of LNSG vs. NNSG (**c**), LNSC vs. NNSC (**d**) and LNSB vs. NNSB (**e**).

**Figure 5 plants-14-02993-f005:**
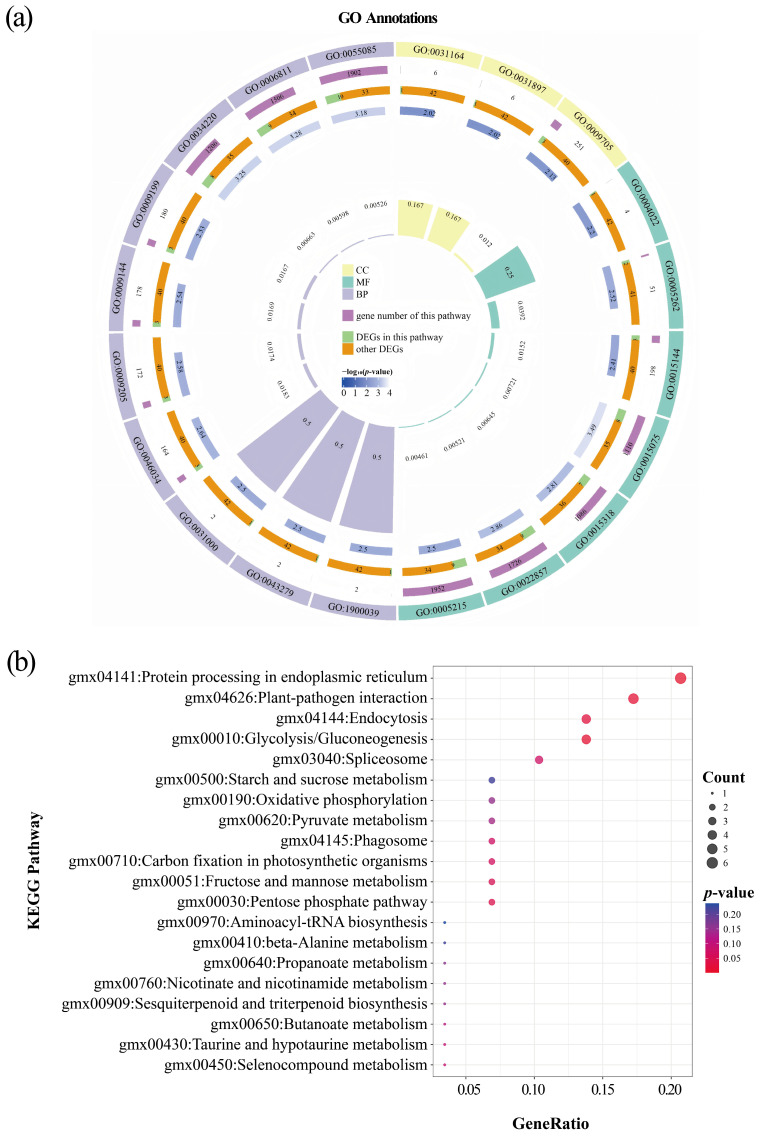
Functional annotations of DEGs between LN and NN treatments at the SG stage (LNSG vs. NNSG). (**a**) GO enrichment circle map of DEGs. Tracks are arranged from outermost to innermost: GO terms, yellow tracks represents cellular component (CC), green tracks represents molecular function (MF), purple tracks represents biological process (BP); BgRatio, the ratio of the number of background genes annotated to a GO term (the length of purple tracks represent the number of genes) to the total number of background genes (26,949); GeneRatio, the ratio of the number of DEGs annotated to a GO term (the green parts in the track) to the total number of DEGs (43); −log_10_(*p*-value), the lighter the color, the more significant it is; the ratio of the number of DEGs annotated to a GO term to the number of background genes annotated to a GO term, the higher the height, the greater the ratio. (**b**) KEGG pathway enrichment bubble plot of DEGs. The x-axis shows the GeneRatio, the y-axis shows the metabolic pathway terms. The size of the plotted circle indicates the number of queried genes. The fill color is scaled by the *p*-value.

**Figure 6 plants-14-02993-f006:**
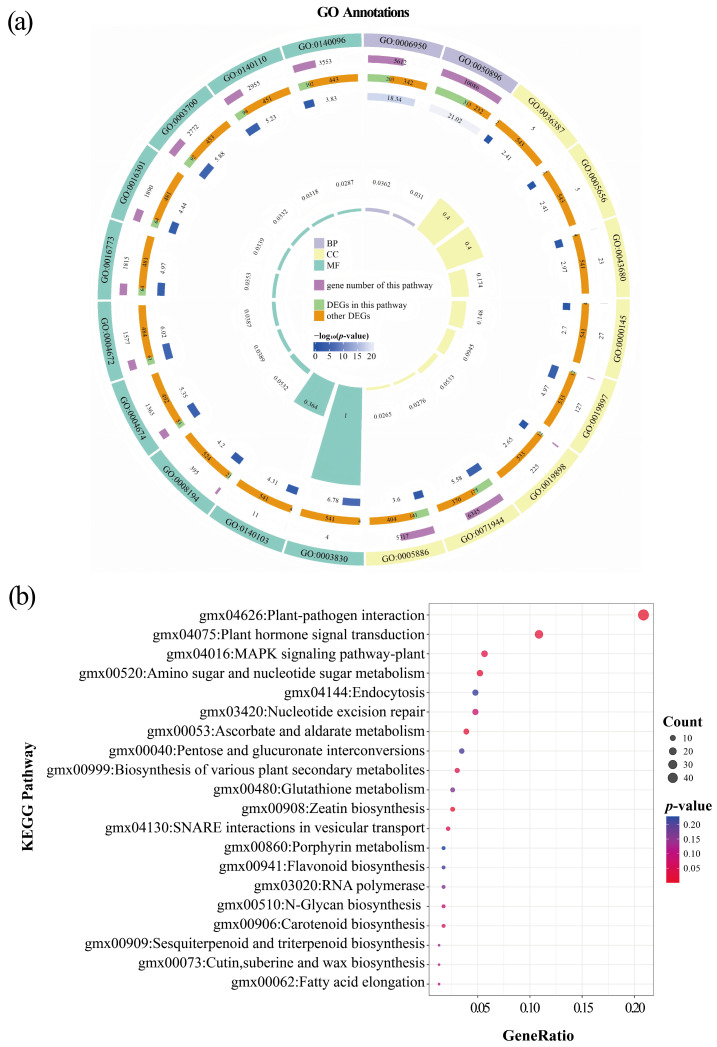
Functional annotations of DEGs between LN and NN treatments at the SC stage (LNSC vs. NNSC). (**a**) GO enrichment circle map of DEGs. (**b**) KEGG pathway enrichment bubble plot of DEGs. The legend is the same as in Figure 5.

**Figure 7 plants-14-02993-f007:**
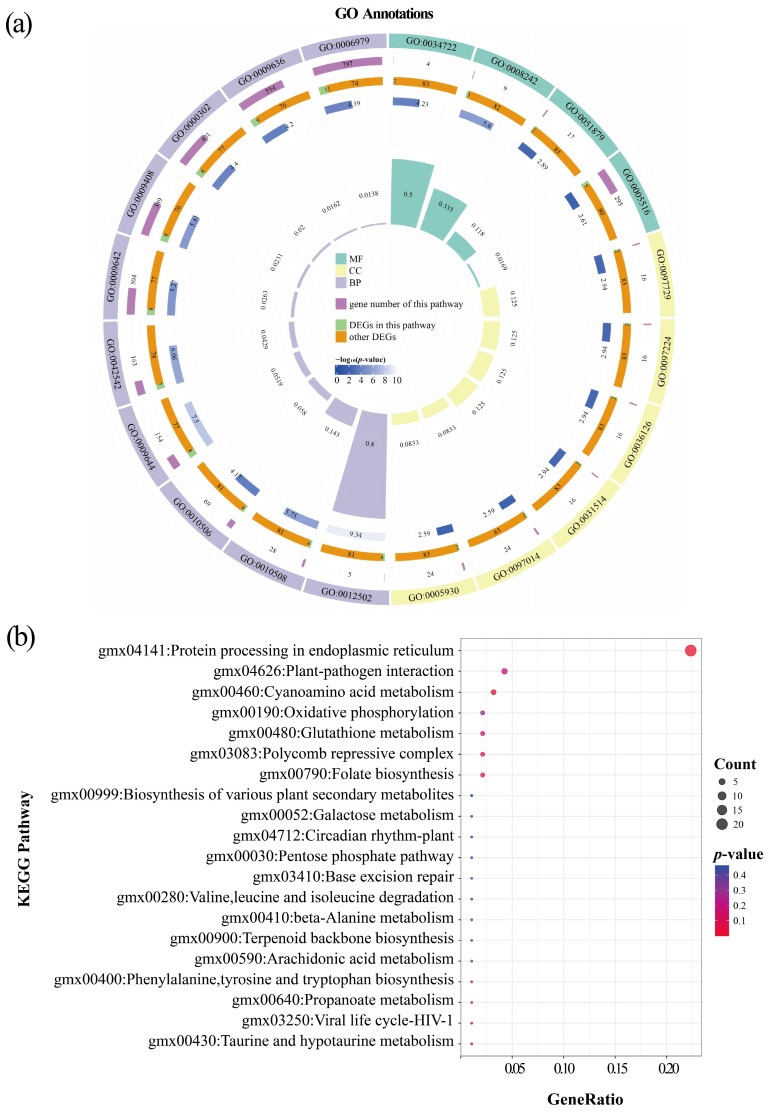
Functional annotations of DEGs between LN and NN treatments at the SB stage (LNSB vs. NNSB). (**a**) GO enrichment circle map of DEGs. (**b**) KEGG pathway enrichment bubble plot of DEGs. The legend is the same as in Figure 5.

**Figure 8 plants-14-02993-f008:**
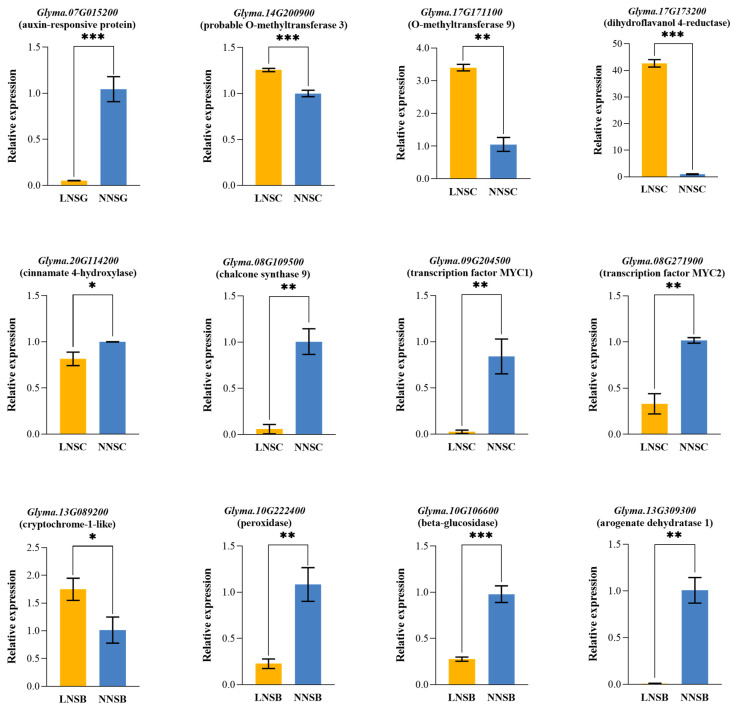
Relative expression levels of DEGs verified by qRT-PCR. *, *p* < 0.05. **, *p* < 0.01. ***, *p* < 0.001.

**Table 1 plants-14-02993-t001:** Candidate genes and annotations.

Gene ID	Annotation
*Glyma.13G309300*	arogenate dehydratase 1
*Glyma.20G114200*	cinnamate 4-hydroxylase
*Glyma.13G089200*	cryptochrome-1-like
*Glyma.17G171100*	O-methyltransferase 9
*Glyma.14G200900*	probable O-methyltransferase 3
*Glyma.08G109500*	chalcone synthase 9
*Glyma.17G173200*	dihydroflavanol 4-reductase
*Glyma.09G204500*	transcription factor MYC1
*Glyma.08G271900*	transcription factor MYC2
*Glyma.07G015200*	auxin-responsive protein
*Glyma.10G222400*	peroxidase
*Glyma.10G106600*	beta-glucosidase

## Data Availability

Data are contained within the article and Appendix A. The raw metabolomic data have been uploaded to the National Genomics Data Center (NGDC), with the corresponding accession number PRJCA046371.

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
