# Peer review of "Integrated Analysis of Transcriptome and Metabolome Reveals the Accumulation of Anthocyanins in Black Soybean (*Glycine max* L.) Seed Coats Induced by Low Nitrogen Concentration in the Nutrient Solution"

_plants, 2025, doi:10.3390/plants14192993_

Round 1
Reviewer 1 Report
Comments and Suggestions for Authors
Major:
1) The authors should significantly improve the legends for the used figures: e.g. all used abbreviations (e.g. Fig. 1, SG, SC, SB, LN, NN).
2) Fig. 1 – increase image quality and include the bar (e.g. 1 cm).
3) How many biological repeats were made for the data shown in Figure 1? Why such a small mistake? Is it SEM or SD?
4) Increase the font size in the Fig. 2, 3, 4c,d,e, 5,6, and 7.
5) Explain the "Low Nitrogen" conditions in detail. Line 763-764: “Prior to the peak flowering stage, JH6 plants were irrigated with 1/2 Hoagland nutrient solution at a nitrogen concentration of 7.5 mmol/L” –
- a) explain “Hoagland nutrient solution”,
- b) “nitrogen concentration of 7.5 mmol/L” - in what chemical compound is nitrogen present in this solution?
- c) How often and to what extent was this solution irrigated to the plant?
Minor:
6) in title – include the Latin name of black soybean (black soybean Glycine max).
7) in title – “Low Nitrogen” correct to “Low Nitrogen Concentration in the Nutrient Solution”.
8) Line 778: “was centrifuged at 12,000 rpm” – what kind of centrifuge, a rotor?
9) Line 782: “cyanidin-3-O-sophoroside” – which manufacturer of the standard?
10) Line 814-815: “and the sequencing analysis was conducted by Jiangsu Sanshu Biotechnology Co., Ltd.” - This information is presented above, line 789.
Comments on the Quality of English LanguageA small edit is needed
Reviewer 2 Report
Comments and Suggestions for Authors
The paper touches interesting and relevant topic. The authors present a theoretical background and new insights into how low-nitrogen treatment affects anthocyanin accumulation in black soybeans. The main advantage of the study is the use of transcriptomic and metabolomic analysis combined with qPCR validation. However, there are some comments to the article:
- Figure 1b. The annotation does not specify whether the average values are presented as mean ± SEM or mean ± SD.
- Figure 1b. SC NN. Why is there no ± SEM or ± SD? Visually, there are no significant differences. Show the mean ± SEM or mean ± SD as a table in additional materials, in the description of the results, or above the columns in Figure 1b.
- Figure 2a. PC1(...%) is missing from the graph.
- Figure 2c. The color scale ranging from blue (−2) to red (+2) is presented without clarification. It is Z-score?
- Figures S1-S3. “group group” in legend.
- Line 206. What does “VIP” mean?
- The figures and tables are arranged in a strange order. They should be placed after their first mention in the text.
- Lines 320-329, 376-377 relate more to materials and methods.
- Figure 4a. What does legend mean? The Pearson correlation coefficient?
- Figure 4e. Is log2(FoldChange) 20 on the X-axis superfluous?
- Figures 5,6,7. Please complete the annotation. What's under (a) and (b)? What does the first inner circle in (a) mean?
- Line 388: “(6 genes, 2 downregulated and 2 upregulated)”. 4 genes?
- Figure 8. To enhance the clarity of the figure, please add the annotation from Table 1 below the gene ID in parentheses.
- Lines 748-749: “Overexpression of PagCRY1 dramatically increased anthocyanin accumulation in Populus”. There is no the reference.
- Line 416. The phrase “Integrated analysis of transcriptomic and metabolomic” appears to be truncated.
- Table 1, Lines 420-421. The term “significant changes” requires clarification. Does it refer to the highest or lowest log2(FoldChange) values? Additionally, it is unclear why the expression analysis of the selected genes was performed only at specific stages rather than across all stages.
- The raw transcriptomic and metabolomic data have not been deposited in public databases. It is essential to make these data publicly available to ensure transparency and reproducibility of the study. Please deposit the raw data in an appropriate public repository (e.g., SRA, Gene Expression Omnibus (GEO), Metabolomics Workbench) and provide the corresponding accession numbers in the manuscript.
- Lines 801-802: “Transcriptome data quality was evaluated using RSeQC software”. Transcriptome quality data is not presented in the results.
- Line 825: “were designated as differential metabolites for subsequent bioinformatics analysis”. Please describe the bioinformatic analysis.
- 4.5. Transcriptomic analysis. The software used to perform the GO and KEGG enrichment analysis is not described.
- Line 797: “second-generation high-throughput sequencing platform”. Please specify the sequencing platform used. Also, provide information about the library preparation kit and method.
Reviewer 3 Report
Comments and Suggestions for Authors
The presented manuscript describes the results of the transcriptomic and metabolomic analysis regarding the anthocyanins accumulation in the seed coats of black beans grown in the conditions of low nitrogen supply.
I need to underline that I have no substantial concerns or comments on the work. In my opinion it has been designed and prepared appropriately presenting consistent methodological approach. The introduction and discussion sections contain some repetitive fragments (as marked in the attached pdf file) but it does not lower the overall merit of these parts. Data presentation was conducted with high clarification.
What can be improved is the proposing in the conclusion or discussion section of some future perspectives or required further studies in the undertaken area as to place the work in a wider perspective.
I also suggest some small modification of the title as in the current form it suggests that the accumulation of anthocyanins was revealed by both transcriptome and metabolome analysis while it was clearly visible phenotypically.
In the methodology section 4.4 RNA extraction, library construction and sequencing - the procedure needs to be described in more details, includign the applied conditions, used kits, equipment etc. Also full reference to all equipment, kits needs to be provided (see attached pdf file).
Some additional minor comment are presented in the attached pdf file.
Overall, in my opinion after the minor corrections the manuscript can be accepted for the publication.

Reviewer 4 Report
Comments and Suggestions for Authors
Synopsis
This manuscript presents an inquiry into the transcriptome and metabolome of the soybean strain Jinda Heidou No. 6 (JH6) grown under control and nitrogen deficient conditions. Three developmental stages of seeds were used, SG (Seed Green ~ 29 days after flowering), SC and SB (black seeds). Anthocyanin levels were assayed in soybean seeds under nitrogen limiting and control values in the three seed developmental stages. The transcriptomes of the seeds was performed to identify Differentially Expressed (DE) genes between normal and limiting nitrogen levels at the three developmental stages (SG,SC,BC) . Pathway analysis was performed on the DE genes using KEGG pathways and GO terms. The manuscript specifically focuses on 12 genes that were “significantly” differentially expressed and already known to be involved in Anthocyanin biosynthesis and autophagy. The expression of these 12 genes were determined by RT-PCR.
Issues
Figure 1 legend.
Panel A. The legend needs to be more descriptive. Specify what SG, SC and SB are. Panel B. The “different treatments” should be specified to make the legend more informative.
The data for figure1 B should be included in the supplemental data section as a table.
Figure 2 legend.
Panel B. The data for this panel should be included in the supplemental data section as a table.
Panel C. The data for this panel should be included in the supplemental data section as a table.
Figure 3 legend
Panel A. The data for this panel should be included in the supplemental data section. Venn is misspelled as “Veen”
Panel B,C and D. The data for these panels should be included in the supplemental data section as separate tables.
Figure 4 legend.
Panel A. The data for this panel should be included in the supplemental data section as a table.
Panel B. The data for this panel should be included in the supplemental data section as a table.
Panels C,D,E. The list and values for the significant genes for these panels should be included in the supplemental data section as a table.
Figure S1.
The data for this figure should be included in the supplemental data section as a table.
Figure S2.
The data for this panel should be included in the supplemental data section as a table.
Figure S3.
The data for this figure should be included in the supplemental data section as a table.
Figure S4.
What chemical was measured? The axis is not properly labelled.
Figure 5.
Panel A. The data presented in the panel is not adequately described. The data for this panel should be included in the supplemental data section as a table. While the totals in each category are interesting, the list of each gene in the category should be presented in the supplementary data.
Panel B. The data presented in the panel is not adequately described. The data for this panel should be included in the supplemental data section as a table.
Figure 6.
Panel A. The data presented in the panel is not adequately described. The data for this panel should be included in the supplemental data section as a table. While the totals in each category are interesting, the list of each gene in the category should be presented in the supplementary data. While the totals in each category are interesting, the list of each gene in the category should be presented in the supplementary data.
Panel B. The data presented in the panel is not adequately described. The data for this panel should be included in the supplemental data section as a table.
Figure 7.
Panel A. The data presented in the panel is not adequately described. The data for this panel should be included in the supplemental data section as a table. While the totals in each category are interesting, the list of each gene in the category should be presented in the supplementary data. While the totals in each category are interesting, the list of each gene in the category should be presented in the supplementary data.
Panel B. The data presented in the panel is not adequately described. The data for this panel should be included in the supplemental data section as a table.
Line 327. What is “rich factor (GeneRatio)”? This needs a description
Lines 734-735. Does autophagy “trigger” anthocyanin biosynthesis or does it “support” it? Since anthocyanins are made in LN and NN conditions, I would say that it “supports” biosynthesis even though the plant is in a LN condition.
Line 799. What options were used in the raw sequencing data filtering with FastP software?
Line 800. What options were used in the STAR software to map the reads to the reference genome?
Line 802. What options were used in the RSeQC software (R package?) to determine the transcriptome assembly quality?
Lines 846-847. Since a single black seeded cultivar (Jinda Heidou No. 6, JH6) was used in this experiment, the inference space is limited to just JH6. Had more black-seeded strains been assayed, then you could say “black soybeans”. This should really be “black-seeded soybeans” since no soybean plants are “black”.
Round 2
Reviewer 1 Report
Comments and Suggestions for Authors
Accept
Comments on the Quality of English LanguageA small edit is needed
Reviewer 4 Report
Comments and Suggestions for Authors
My criticisms were addressed.